# Retinal Cyclic Nucleotide-Gated Channel Regulation by Calmodulin

**DOI:** 10.3390/ijms232214143

**Published:** 2022-11-16

**Authors:** Aritra Bej, James B. Ames

**Affiliations:** Department of Chemistry, University of California, Davis, CA 95616, USA

**Keywords:** CNGA1, CNGB1, photoreceptor, retina, calmodulin, phototransduction

## Abstract

Retinal cyclic nucleotide-gated (CNG) ion channels bind to intracellular cGMP and mediate visual phototransduction in photoreceptor rod and cone cells. Retinal rod CNG channels form hetero-tetramers comprised of three CNGA1 and one CNGB1 protein subunits. Cone CNG channels are similar tetramers consisting of three CNGA3 and one CNGB3 subunits. Calmodulin (CaM) binds to two distinct sites (CaM1: residues 565–587 and CaM2: residues 1120–1147) within the cytosolic domains of rod CNGB1. The binding of Ca^2+^-bound CaM to CNGB1 promotes the Ca^2+^-induced desensitization of CNG channels in retinal rods that may be important for photoreceptor light adaptation. Mutations that affect Ca^2+^-dependent CNG channel function are responsible for inherited forms of blindness. In this review, we propose structural models of the rod CNG channel bound to CaM that suggest how CaM might cause channel desensitization and how dysregulation of the channel may lead to retinal disease.

## 1. Overview of Retinal CNG Channels

Cyclic nucleotide-gated channels [1,2] were first identified in the plasma membrane of vertebrate rod photoreceptor cells, where they serve a functional role in visual phototransduction [3]. CNG channels are cation-specific channels that are opened by the direct binding of intracellular cyclic nucleotides [4]. CNG channels consist of multiple protein subunits coded by six different genes grouped into two subtypes: CNGA and CNGB, as shown in Figure 1A [5]. CNGA1 from the bovine retina was the first gene to be cloned [6] and was originally called the α-subunit of the retinal rod CNG channel. CNGA1 is mainly expressed in rod photoreceptors and recombinant CNGA1 can be expressed as a functional homomeric channel that can be gated by cGMP [7,8]. Mutations in CNGA1 are genetically linked to a recessive form of retinitis pigmentosa [9]. The native rod CNG channel contains a second protein subunit, CNGB1, [10] originally called the β-subunit. The native rod CNG channel forms a hetero-tetramer composed of three CNGA1 and one CNGB1 subunits [11], as seen in recent atomic-level structures of the rod CNG channel [12,13,14]. The CNGA1 and CNGB1 structures (Figure 1B) are both comprised of an N-terminal cytosolic region (not visible in the cryo-EM), six transmembrane spanning helices (channel domain, CD) and an intracellular cGMP-binding domain (CNBD) flanked by a C-terminal leucine zipper domain (CLZ) in CNGA1 or a CaM-binding domain (CaM2) in CNGB1. The sixth transmembrane helix (S6) from each subunit forms the ion-conducting pathway (or channel pore). The CNBD is comprised of three helices and a β-roll containing eight β-strands. An 80-residue linker (C-linker) between the CNBD and the channel domain relays ligand-induced conformational changes to the S6 helix in the channel pore [15]. 

Cone photoreceptors express CNG channels that contain two different protein subunits, CNGA3 (cone α subunit) and CNGB3 (cone β subunit) [16,17] that are distinct from the protein subunits of the rod channel. Recombinant CNGA3 subunits expressed in non-native cells form functional homomeric channels that are gated by cGMP [16,18,19], similar to that seen for CNGA1 [8]. The native cone CNG channel forms a hetero-tetramer composed of three CNGA3 and one CNGB3 subunits, as seen in the recent cryo-EM structure [20]. Mutations in human CNGA3 and CNGB3 have been linked to achromatopsia (also called color blindness), which is an autosomal recessive congenital disorder characterized by the complete inability to discriminate between colors [21,22].

**Figure 1 ijms-23-14143-f001:**
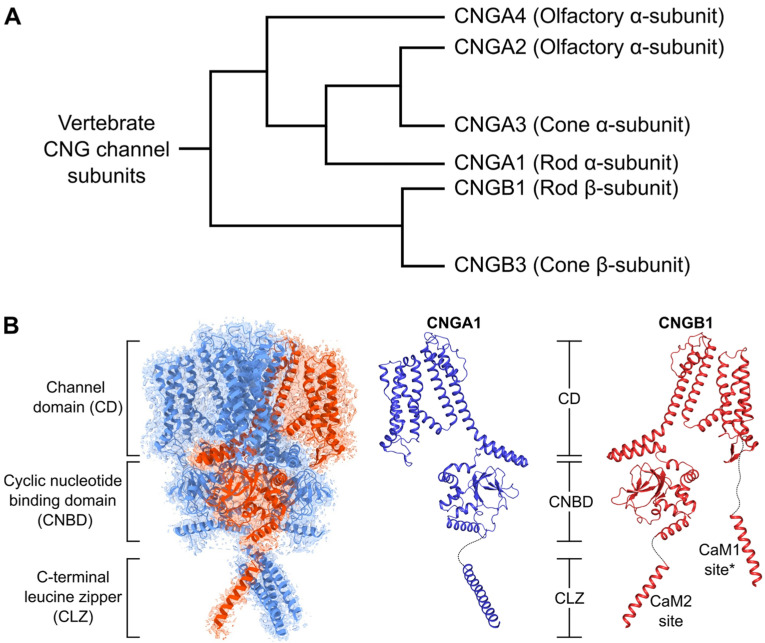
Structural overview of CNG channels. (**A**) Phylogenetic analysis of CNG channel subunits. (**B**) Cryo-EM structure of CNG channel [14] and ribbon diagrams of CNGA1 (blue) and CNGB1 (red) subunits. The structure of the CaM1 site (asterisks) was not visible in the cryo-EM structures and instead was modeled by AlphaFold [23].

## 2. Retinal CNG Channels Are Essential for Visual Phototransduction

Phototransduction in retinal rod photoreceptor cells is triggered by light-activation of the visual pigment rhodopsin (R*) that turns on an enzymatic cascade, leading to the hydrolysis of cGMP catalyzed by phosphodiesterase PDE6 [24,25,26,27] (Figure 2A). The cytosolic cGMP concentration is maintained at a high level in dark-adapted photoreceptors, which allows for CNG channels to remain open in the absence of light [26]. Photoreceptor CNG channels conduct a cationic current only when the cytosolic ligand-binding domain binds to intracellular cGMP (Figure 2B) [3,24]. The light-induced lowering of cytosolic cGMP (that occurs during phototransduction) causes the closure of CNG channels in the photoreceptor’s outer-segment plasma membrane that blocks the entry of Na^+^ and Ca^2+^ and generates a neural signal [28,29,30]. The light-induced CNG channel closure during phototransduction causes a decrease in the cytosolic Ca^2+^ concentration (500 nM in the dark [31] versus 50 nM in the light [32]) that, in turn, modulates CNG channel sensitivity [19,33,34] and activates guanylate cyclase [35,36,37]. CNG channels have the highest open probability at low cytosolic Ca^2+^ levels in light-activated photoreceptors and the channels become desensitized at elevated Ca^2+^ levels that exist in dark-adapted photoreceptors ([Ca^2+^]_i_ = 500 nM [31]). The Ca^2+^-dependent desensitization of photoreceptor CNG channel activity is mediated by CaM, which is important for promoting light adaptation in rod photoreceptor cells [33,34,38,39]. The Ca^2+^-dependent CNG channel activity might be due to a Ca^2+^-induced lowering of the cGMP-binding affinity [19,40,41]. Ca^2+^-dependent CNG channel activity has also been reported in the cone photoreceptor [42,43,44,45]. 

## 3. Calmodulin Mediates Ca^2+^-Dependent CNG Channel Desensitization

Calmodulin (CaM) is a 16.7 kDa Ca^2+^ sensor protein that belongs to the EF-hand superfamily [46]. CaM contains four EF-hand Ca^2+^-binding motifs (EF1, EF2, EF3 and EF4) that form two separate domains (EF1 and EF2 form the CaM N-lobe, while EF3 and EF4 form the CaM C-lobe) [47]. The CaM C-lobe and N-lobe each bind to Ca^2+^ with a dissociation constant of ~1 μM and 10 μM, respectively [48], which allows for Ca^2+^ to bind to the C-lobe first before binding to the N-lobe. The Ca^2+^-saturated form of CaM binds to hundreds of different target proteins, including dozens of enzymes, receptors, ion channels and other Ca^2+^ transporters [49]. 

The Ca^2+^-dependent desensitization of rod CNG channels requires CaM binding to two separate sites within the cytosolic region of CNGB1 (residues 565–587 called CaM1 [7,19,41,50] and residues 1120–1147 called CaM2 [41] in Figure 3A). The CaM1 and CaM2 sites are both conserved in the cone CNGB3 (Figure 3A) and the cone CNG channel undergoes Ca^2+^-induced desensitization [42]. The Ca^2+^-dependent activity of cone channels may also be mediated by a novel Ca^2+^ sensor protein called CNG modulin [43,51]; however, more recent studies suggest that CNG modulin does not regulate CNG channels [52]. The N-terminal CaM1 site in the rod channel was originally identified to bind to Ca^2+^-bound CaM [41] (but not Ca^2+^-free CaM [53]) and CaM binding was suggested to inactivate rod CNG channel activity [19,54]. The CaM1 site (in the absence of CaM) was proposed to interact with a C-terminal domain in CNGA1, known as a C-terminal leucine zipper (CLZ) [11], that is immediately adjacent to the cyclic-nucleotide binding domain (Figure 1B). The binding of Ca^2+^-bound CaM to CaM1 was suggested to block CaM1 binding to the CLZ [7,11]. The N-terminal CaM1 site is not visible in the recent cryo-EM structures [12,13,14,15], suggesting that CaM1 is dynamically disordered or otherwise not visible in these structures. A recent NMR structure of CaM bound to a peptide fragment of CaM1 [53] reveals that the Ca^2+^-bound CaM N-lobe binds stably to CaM1 (Figure 3B), but the Ca^2+^-bound CaM C-lobe binds to CaM1 with much lower affinity, which prevents determination of its structure [53]. Thus, CaM1 binds functionally to only the CaM N-lobe, while the lower affinity CaM C-lobe binding may not be physiologically relevant. The CaM1 residue F575 makes extensive hydrophobic contacts with the Ca^2+^-bound CaM N-lobe (Figure 3B) and the mutation F575E can selectively disable the CaM N-lobe binding to CaM1 [53]. Future electrophysiology studies are needed to study the effect of the F575E mutation on Ca^2+^-dependent desensitization of CNG channels to test whether CaM1 is essential for CNG channel function.

A closer look at the cryo-EM structure of the rod CNG channel in the apo state [13] revealed faint electron density from CaM that is bound to a site in CNGB1 downstream of the CNBD, dubbed the D-helix (Figure 3C). The structure of CaM bound to the D-helix (Figure 3C) is very similar to the crystal structure of the CaM C-lobe bound to the creatine kinase peptide [55]. Moreover, the CaM/D-helix structure (Figure 3C) is essentially identical (within experimental error) to the very recent NMR structure of the CaM C-lobe bound to the CaM2 peptide (Figure 3D) [53]. Indeed, the CaM C-lobe alone can bind to the CaM2 peptide with nanomolar affinity, in contrast to the CaM N-lobe that binds to CaM2 with lower affinity [53]. The nanomolar binding of the CaM C-lobe to the CaM2 peptide explains in part how the CaM/CaM2 complex can bind to Ca^2+^ at the very low Ca^2+^ concentration (80 nM) used in the cryo-EM study [13]. The NMR structure of the Ca^2+^-bound CaM C-lobe bound to the CaM2 peptide (Figure 3D) reveals CaM2 residues L1132 and L1136 make extensive hydrophobic contact with the CaM C-lobe (Figure 3D), and the mutations (L1132E and L1136E) selectively disable CaM C-lobe binding to the CaM2 peptide [53]. Future electrophysiology studies are needed to study the effect of the L1132E/L1136E mutations on the Ca^2+^-dependent desensitization of CNG channels and to determine whether CaM binding to the CaM2 site is essential for CNG channel function. 

## 4. Structural Models of CNG Channel Regulation by Calmodulin

### 4.1. Two-Site Model for CaM Regulation of the Rod CNG Channel

We propose two different structural models of Ca^2+^-dependent desensitization of CNG channels mediated by CaM: one-site versus two-site model (Figure 4A,B). The two-site model (Figure 4A), originally proposed by A. Bej and J.B. Ames [53], suggests that two separate CaM binding sites (CaM1 and CaM2 in Figure 3A) in CNGB1 bind selectively to each lobe of CaM: CaM1 binds to the CaM N-lobe and CaM2 binds to the CaM C-lobe. At high-cytosolic Ca^2+^ levels in the dark-adapted photoreceptor, we propose that a single CaM bridges the two CaM binding sites together to bring CaM1 and CaM2 in close proximity in the desensitized-channel state (Figure 4A, left panel). The Ca^2+^-induced bridging of CaM1 and CaM2 and their close proximity with the CLZ may somehow alter the structure of the adjacent CNBD to decrease the binding affinity of cGMP or otherwise reduce the channel open probability. By contrast, the low cytosolic Ca^2+^ level in the light-activated photoreceptor should cause Ca^2+^ dissociation from the CaM N-lobe, which is predicted to disable CaM N-lobe binding to CaM1 and, therefore, allow for CaM1 and CaM2 to move far apart when the channel switches to the highest open probability state (Figure 4A, right panel). In essence, the CaM N-lobe serves as a Ca^2+^ sensor, while the Ca^2+^-bound CaM C-lobe is proposed to remain constitutively anchored to CaM2 even at low Ca^2+^ levels in light-activated rods, because the apparent Ca^2+^ affinity of CaM bound to CaM2 is estimated to be ~60 nM [53]. This pre-anchoring of the Ca^2+^-bound CaM C-lobe to the sensitized CNG channel at low Ca^2+^ levels eliminates any diffusional barrier of Ca^2+^ sensing and is reminiscent of CaM pre-association with L-type voltage-gate Ca^2+^ channels [56,57]. The two-site model predicts that Ca^2+^ binding to the CaM C-lobe (EF3 and EF4) should be essential for anchoring the CaM C-lobe to the sensitized CNG channel (Figure 4A, right panel). Future experiments on the CaM mutant, CaM_34_ (that disables Ca^2+^ binding to the CaM C-lobe [58]) are needed to test whether Ca^2+^-binding to the CaM C-lobe is essential for CNG channel sensitization at low Ca^2+^ levels in light-activated photoreceptors. Another way to test the two-site model would be to assess the effect of CNG channel mutations F575E and L1132E/L1136E on the channel’s open probability and cGMP binding affinity. The two-site model (Figure 4A) predicts that the F575E mutation should disable CaM N-lobe binding, which should prevent Ca^2+^-induced channel desensitization and/or permit high-affinity cGMP binding regardless of the Ca^2+^ level. The L1132E/L1136E mutations should disable CaM binding to CaM2, which, in turn, might destabilize the sensitized channel state, causing a decrease in the channel open probability and perhaps a lower cGMP binding affinity. Future electrophysiology experiments on the F575E and L1132E/L1136E mutations are needed to test the predictions of the two-site model.

### 4.2. CaM Binding to CaM2 May Be Sufficient for CNG Channel Desensitization

A one-site model for Ca^2+^-induced CNG channel desensitization is proposed, in which CaM C-lobe binding to CaM2 alone may be sufficient to cause desensitization (Figure 4B). The CaM interaction with CaM1 is not considered here because the CaM1 peptide has relatively low affinity binding to the CaM N-lobe (K_d_ = 5 μM [53]), which may be outside the physiological range of cellular-free CaM concentration (100 nM [60]). This is in stark contrast to the CaM2 site that binds to CaM with a K_d_ in the nanomolar range [53]. Moreover, the CaM N-lobe and CaM1 site are both structurally disordered or otherwise not visible in the cryo-EM structures [12,13,14,15], consistent with a lack of CaM1 function. We propose that the partially open channel state described previously in [13,14,15] with CaM bound to CaM2 (Figure 4B) might represent the desensitized CNG channel state in dark-adapted photoreceptors that contain high-cytosolic Ca^2+^ and cGMP levels. A recent cryo-EM structure of the apo CNG channel revealed that the CaM C-lobe is bound to CaM2 (Figure 3C). Our one-site model proposes that CaM binding to CaM2 may stabilize the CNGB1 subunit in a desensitized conformation in which CNGB1 residues in the S6 helix (residues F872, I876 and R880) point inward to block the ion conduction pathway [13] (Figure 4B, inset). This channel pore blockage by CNGB1 residues (F872, I576 and R880) was previously termed the “CNGB1 gate” [15] (Figure 4C). In essence, CaM binding to CaM2 is proposed to keep CNGB1 in a closed conformation by closing the CNGB1 gate (Figure 4C, left and middle panels). Therefore, CaM binding to CaM2 may be necessary to stabilize the partially open channel state (Figure 4C, middle panel) that we propose may represent the desensitized channel in dark-adapted photoreceptors (Figure 4B). We furthermore suggest that the low-cytosolic Ca^2+^ level in the light-activated photoreceptor ([Ca^2+^]_i_ ~50 nM) might cause at least some of the bound Ca^2+^ to dissociate from the CaM C-lobe (K_d_ for Ca^2+^ binding to CaM C-lobe bound to CaM2 is 60–80 nM [53]), which, in turn, might cause Ca^2+^-free CaM to dissociate from CaM2 and possibly allow the CaM-free channel to convert into a more fully open state (Figure 4C, right panel) akin to the fully open state of the homomeric CNGA1 channel [59]. In summary, we propose that Ca^2+^-induced CNG channel desensitization occurs when the fully open channel (at low Ca^2+^ levels) converts into the partially open state stabilized by CaM binding to CaM2 at high Ca^2+^ levels. This one-site model (Figure 4B) predicts that the L1132E/L1136E mutations (that disable CaM binding to CaM2 [53]) should abolish Ca^2+^-dependent channel desensitization. An important difference between the one-site and two-site models (Figure 4A,B) is that CaM1 should be essential for Ca^2+^-induced CNG channel desensitization in the two-site model, but not in the one-site model. Another difference is that CaM binding to CaM2 is predicted to stabilize the partially open (desensitized) state in the one-site model (Figure 4B), in contrast to possibly stabilizing the high open-probability state in the two-site model (Figure 4A, right panel). Future electrophysiology experiments and cryo-EM structures of CNG channels bound to CaM (at both high and low Ca^2+^ levels) are needed to discriminate the two models and test their predictions. 

## 5. Conclusions

Retinal rod CNG channels exhibit Ca^2+^-induced channel desensitization caused by CaM that may be important for light adaptation. CaM binding to CNG channels is proposed to decrease the channel open probability and/or decrease its cGMP binding affinity at high-cytosolic Ca^2+^ levels in dark-adapted photoreceptors. A cryo-EM structure of the apo CNG channel [13] suggests that CaM is bound to a CaM binding site (helix D) in CNGB1 that interacts with the CLZ domain. A recent NMR structural analysis [53] suggests that helix D in CNGB1 may be the C-terminal CaM site (CaM2). The NMR structures of CaM bound to separate CNGB1 sites (CaM1 and CaM2) predict that a single CaM could bind to each CNG channel tetramer by having the CaM C-lobe bind to the CNGB1 C-terminal site (CaM2) and the CaM N-lobe bind to the N-terminal site (CaM1). We propose two possible models to explain the Ca^2+^-induced desensitization of CNG channels caused by CaM. A two-site model (Figure 4A) proposes that a single CaM binds to two separate sites in CNGB1 (CaM1 and CaM2) to cause channel desensitization. The separate binding of the CaM lobes to CaM1 and CaM2 brings the two sites in close proximity to possibly cause channel inactivation at high Ca^2+^ levels. An alternative one-site model (Figure 4B) proposes that CaM binding to CaM2 alone may be sufficient to cause channel desensitization. An important difference between the two models is that the CaM1 site is essential for Ca^2+^-induced CNG channel desensitization in the two-site model (Figure 4A), but not in the one-site model (Figure 4B). Moreover, CaM binding to CaM2 is predicted to increase the channel open probability in the two-site model, in contrast to promoting channel desensitization in the one-site model. Future electrophysiology experiments and cryo-EM structures of CNG channels bound to CaM are needed to test the two models. Mutations that affect the Ca^2+^-dependent desensitization of CNG channels are linked to inherited forms of blindness. Understanding the structural basis of CaM binding to the CNG channel may provide new insights for the future design of therapeutics.

## Figures and Tables

**Figure 2 ijms-23-14143-f002:**
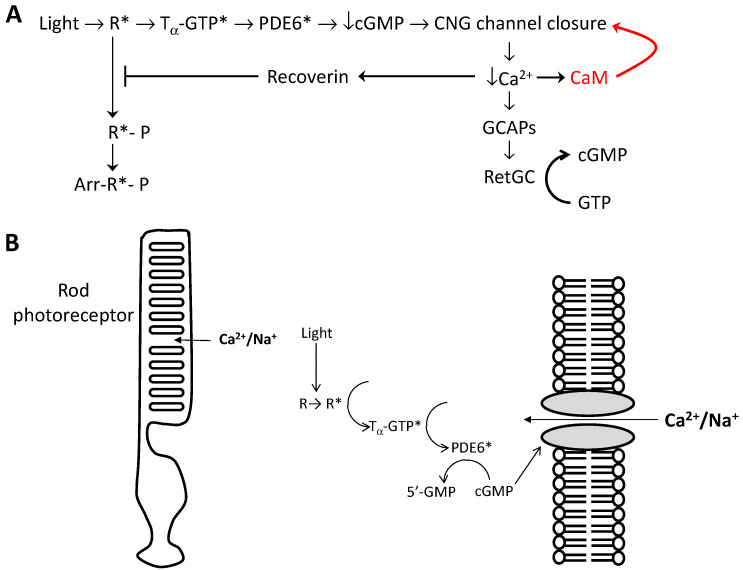
CNG channel function in visual phototransduction. (**A**) Schematic model of visual excitation regulated by intracellular Ca^2+^: Light-excited rhodopsin (R*), GTP-bound transducin (T_α_-GTP), phosphodiesterase (PDE6), retinal guanylate cyclase (RetGC), guanylate cyclase activating proteins (GCAPs), phosphorylated rhodopsin (R*-P), arrestin (Arr). (**B**) CNG channel gating by intracellular cGMP in the plasma membrane of rod photoreceptors.

**Figure 3 ijms-23-14143-f003:**
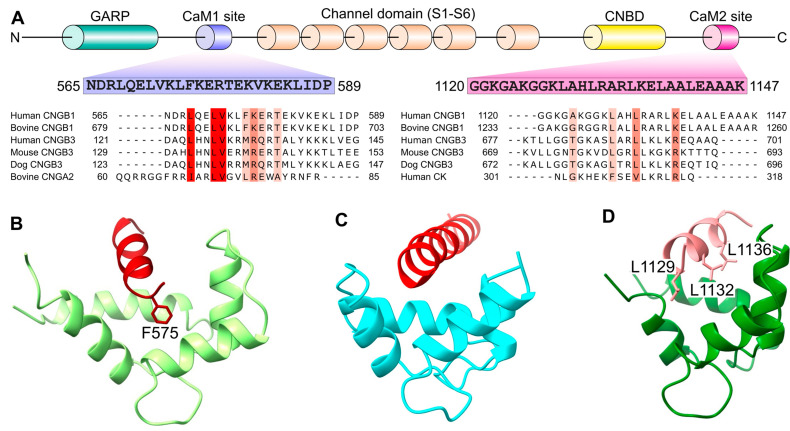
CaM binding sites in rod CNGB1 vs. cone CNGB3. (**A**) Domain architecture and amino acid sequence of CaM binding sites (CaM1 and CaM2) in CNGB1. Sequence alignment of CaM1 and CaM2 with corresponding residues in CNGA2, creatine kinase (CK), cone CNGB3 and vertebrate homologs of CNGB1. Conserved residues are highlighted in color. (**B**) NMR structure of the Ca^2+^-bound CaM N-lobe (light green) bound to CaM1 peptide (red) [53]. (**C**) Cryo-EM structure of CaM C-lobe (cyan) bound to the D-helix of CNGB1 (red) [13]. (**D**) NMR structure of the Ca^2+^-bound CaM C-lobe (dark green) bound to CaM2 peptide (pink) [53].

**Figure 4 ijms-23-14143-f004:**
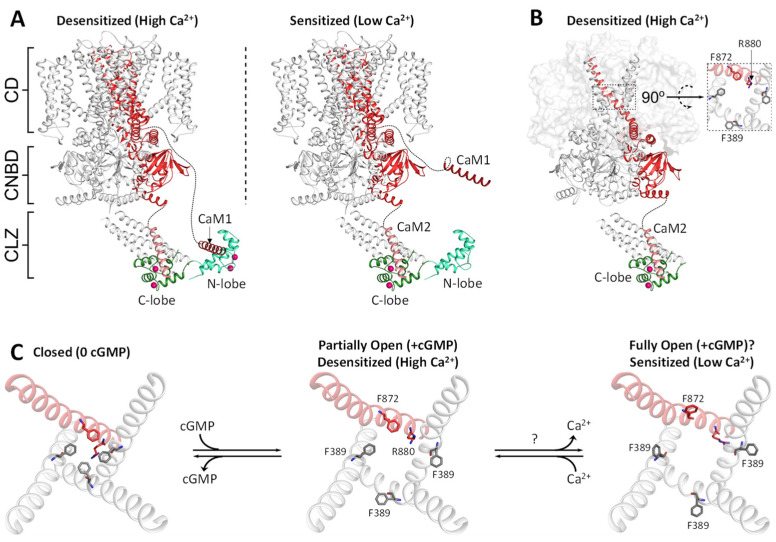
Two structural models of CNG channel regulation controlled by CaM. (**A**) Two-site model proposes that CaM binds to two separate sites (CaM1 and CaM2). Adapted from [53]. CNGB1 is highlighted in red. Bound Ca^2+^ are magenta spheres. The CNG channel at low Ca^2+^ concentration in light-activated photoreceptors has the Ca^2+^-bound CaM C-lobe (dark green) bound to CaM2 (light red) and Ca^2+^-free CaM N-lobe (light green) does not bind to the channel (right panel). The CNG channel at high Ca^2+^ concentration in the dark-adapted photoreceptor has the Ca^2+^-bound CaM C-lobe and N-lobe bound to CaM2 and CaM1, respectively (left panel). (**B**) A one-site model proposes that CaM binds functionally to only one site (CaM2). At high Ca^2+^ levels, the Ca^2+^-bound CaM C-lobe binds to CaM2 and is proposed to stabilize the structure of the S6 helix that causes CNGB1 residues (F872, I876 and R880) to point inward to block the channel pore in the desensitized state (see inset). At low Ca^2+^ levels, the Ca^2+^-free CaM dissociates from CaM2, which could allow for the channel to switch to a proposed fully open state, as seen in the homomeric CNGA1 [59]. (**C**) Proposed model for Ca^2+^-dependent gating of the rod CNG channel. The closed apo state (in the light-activated photoreceptor) has CNGB1 gate residues (L872, I876 and R880) and CNGA1 residues (F389 and V393), each pointing inward to block the ion conduction pathway (left panel). The binding of cGMP causes the formation of a partially open channel (middle panel) in which CNGB1 gate residues are pointing inward to partially block the ion conduction pathway, while CNGA2 residues (F389 and V393) are pointing outward. We propose that the partial open state (middle panel) may represent the desensitized channel state in dark-adapted photoreceptors (high Ca^2+^ and cGMP). A hypothetical fully open channel (right panel) is proposed to exist at low Ca^2+^ levels, in which the CNGB1 gate residues are pointing outward to fully unblock the ion conduction pathway. A similar fully open state was observed for the homomeric CNGA1 channel [59] that has each CNGA1 gate residues pointing outward to fully unblock the ion conduction pathway.

## Data Availability

Not applicable.

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
