# Peer review of "Retinal Cyclic Nucleotide-Gated Channel Regulation by Calmodulin"

_ijms, 2022, doi:10.3390/ijms232214143_

Round 1

Reviewer 1 Report

The review is devoted to the functioning of retinal cyclic nucleotide-gated (CNG) ion channels in photoreceptor rod and cone cells and to the role of calmodulin in their functioning. The authors proposed two possible models to explain Ca2+-induced desensitization of CNG channels caused by calmodulin. The review is well written and well illustrated with structural figures. It will be useful for researchers who study phototransduction, ion channels, and calcium binding proteins. The review is definitely worth publishing. It might be worth adding the review some brief information about the structure and functions of calmodulin, despite the fact that it is, of course, a well-known calcium sensor protein.

Author Response

A brief review of calmodulin is now provided in section 3 on page 3 (lines 84-91).

Reviewer 2 Report

The authors have undertaken a fairly comprehensive review of structural models of the rod cyclic nucleotide-gated channel bound to calmodulin. The authors introduced overview of retinal cyclic nucleotide-gated channels that are essential for visual phototransduction and summarized the mechanisms of Ca2+-dependent desensitization of cyclic nucleotide-gated channel. The authors proposed two possible structural models to explain Ca2+-induced desensitization of cyclic nucleotide-gated channels caused by calmodulin, that is, one-site versus two-site models, and concluded that future electrophysiology experiments and cryo-EM structures of cyclic nucleotide-gated channels bound to calmodulin are needed to test the two models and the structural basis of calmodulin binding to the cyclic nucleotide-gated channel may provide new insights for the design of therapeutics. The manuscript is well-written. I don’t have any concerns.

Author Response

We thank the reviewer for the kind praise and the reviewer does not raise any concerns.

Reviewer 3 Report

This review by Arita Bej et al describes the structural aspects of retinal cyclic nucleotide-gated (CNG) ion channels including cone and rod CNG channels. The review is well organized and the question is well defined addressing 1) a structural overview of rod and cone CNG channels, and 2) the mechanism of visual transduction with specific emphasis given to the calmodulin and Ca2+  dependent desensitization of the CNG channel. Finally, the authors have discussed two structural models predicting the open and closed conformation of the CNG channels in response to the calcium ions under light and dark-adapted conditions.

I have no further suggestions to make and the manuscript is acceptable.

Author Response

The reviewer has no suggestions or points to address.